



# Identifying the spatiotemporal variations of ozone formation regimes across China from 2005 to 2019 based on polynomial simulation and causality analysis

Ruiyuan Li[1], Miaoqing Xu[1], Manchun Li[2], Ziyue Chen[1], Bingbo Gao[3], Na Zhao[4,5], Qi Yao[1]

[1]State Key Laboratory of Remote Sensing Science, College of Global Change and Earth System Sciences, Beijing Normal University, Beijing, 100875, China
[2]School of Geography and Ocean Science, Nanjing University, Nanjing, 210023, China.
[3]College of Land Science and Technology, China Agriculture University, Beijing, 100083, China.
[4]State Key Laboratory of Resources and Environment Information System, Institute of Geographic Sciences and Natural Resources Research, Chinese Academy of Sciences, Beijing, 100049, China
[5]University of Chinese Academy of Sciences, Beijing, 100080, China

*Correspondence to*: Ziyue Chen (zychen@bnu.edu.cn)

**Abstract.** Ozone formation regimes are closely related to the ratio of VOCs to $NO_x$. Different ranges of $HCHO/NO_2$ indicate three formation regimes, including VOCs-limited, transitional and $NO_x$-limited regimes. Due to the unstable interactions between a diversity of precursors, the range of transitional regime, which plays a key role in identifying ozone formation regimes, remains unclear. To overcome the uncertainties from single models and the lack of reference data, we employed two models, polynomial simulation and Convergent Cross Mapping (CCM), to identify the ranges of $HCHO/NO_2$ across China based on ground observations and remote sensing datasets. The ranges of transitional regime estimated by polynomial simulation and CCM were [1.0, 1.9] and [1.0, 1.8]. Since 2013, ozone formation regime has changed to the transitional and $NO_x$-limited regime all over China, indicating ozone concentrations across China were mainly controlled by $NO_x$. However, despite the $NO_2$ concentrations, HCHO concentrations continuously exert a positive influence on ozone concentrations under transitional and $NO_x$-limited regimes. Under the circumstance of national $NO_x$-reduction policies, the increase of VOCs became the major driver for the soaring ozone pollution across China. For an effective management of ozone pollution across China, the emission-reduction of VOCs and $NO_x$ should be equally considered.

## 1 Introduction

With the significant improvement of $PM_{2.5}$ pollution, surface ozone has become a major airborne pollutant across China since 2017 (Li et al., 2019a; Lu et al., 2020). Due to its severe threat to public health even during a short-period exposure, ozone pollution has received growing emphasis from governments and scholars (Liu et al., 2018a; Xie et al., 2019). In the past several years, spatiotemporal distribution of ozone concentrations (Wu and Xie, 2017; Shen et al., 2019a), and the influence of meteorological conditions (Chen et al., 2019b; Cheng et al., 2019; Chen et al., 2020) and anthropogenic


emissions (Cheng et al., 2018; Li et al., 2019a; Li et al., 2020) on ozone concentrations have been massively studied. However, due to the highly complicated ozone formation regime, effective ozone control remains challenging.

Different from $PM_{2.5}$, whose main precursor are $NO_x$, VOCs and $SO_2$, the formation and decomposition of ozone are closely related to two types of precursors, VOCs and $NO_x$. There is a diversity of reactions between VOCs and $NO_x$ under different

meteorological conditions and concentration scenarios (Wang et al., 2017). Since VOCs and $NO_x$ can either promote or restrict ozone production, VOCs/$NO_x$ is crucial for surface ozone concentrations. However, the thresholds at which VOCs/$NO_x$ may promote or restrict ozone production remain unclear (Jin et al., 2017; Schroeder et al., 2017). For instance, under a specific VOCs/$NO_x$ scenario, the reduction of $NO_x$ may conversely increase surface ozone concentrations (Sillman et al., 1990; Kleinman, 1994). Furthermore, given the large variations of meteorological conditions and ozone level across

China, the effects of VOCs/$NO_x$ on surface ozone concentrations also demonstrate notable spatiotemporal patterns. In this case, a comprehensive understanding of how the variations of VOCs and $NO_x$ could influence ozone concentrations under different VOCs/$NO_x$ circumstances is crucial for setting effective emission-reduction policies accordingly in different regions.

To examine the complicated non-linear relationship between ozone concentrations and multiple precursors, a large body of

studies has been conducted (Duncan et al., 2010; Choi et al., 2012; Pusede and Cohen, 2012; Chang et al., 2016; Jin et al., 2020). Through small-scale experiments, $NO_2$ and HCHO proved to be effective proxies for $NO_x$ and VOCs (Sillman et al., 1990; Martin et al., 2004). Since $NO_2$ and HCHO can be monitored using remote sensing data, the two precursors have been increasingly considered in ozone-precursor sensitivity research (Jin et al., 2020; Zhang et al., 2020). Cheng et al. (2018) proved that $NO_2/NO$ presented a good consistence with long-term ozone concentrations in Beijing. However, NO was not an

easily recordable precursor based on satellite observations and not applicable in large-scale monitoring. Cheng et al. (2019) suggested that satellite retrieved HCHO/$NO_2$ was strongly correlated with surface ozone concentrations in Beijing. Different HCHO/$NO_2$ indicates distinct ozone formation regimes, including VOC-limited, transitional and $NO_x$-limited regimes. For VOC-limited ($NO_x$-saturated) regime, the control of VOCs emissions leads to the reduction of organic radicals ($RO_2$), the $RO_2$-$NO_x$ reactions and thus ozone concentrations (Milford et al., 1989). On the contrary, the decrease of $NO_x$ promotes

VOCs-CO reaction, leading to the increase of ozone concentration (Kleinman, 1994). For $NO_x$-limited regime, the reduction of $NO_x$ slows down $NO_2$ photolysis, which products free oxygen atoms for ozone formation, and reduces ozone concentrations. The variations of VOCs exert limited influences on ozone concentrations for this regime (Kleinman, 1994). For transitional (VOCs-$NO_x$ mixed) regime, both VOCs and $NO_x$ impose positive influences on ozone concentrations. Since the transitional regime divides VOC-limited and $NO_x$-limited regimes, the estimation of the transitional regime range plays a

key role to identify different ozone formation regimes.

Duncan et al. (2010) calculated the transitional regime range as [1.0, 2.0] using the Community Multiscale Air Quality Modeling System (CMAQ) model, whose uncertainties may influence the estimation accuracy (Schroeder et al., 2017). Jin et al. (2020) employed a polynomial model and calculated the transitional regime range over U.S. urban areas as [3.2, 4.1] based on decades of remote sensing and ground observation data. However, given the notable difference of meteorological





conditions, ozone levels and the composition of precursors across different countries, whether the transitional regime range
      extracted in US is applicable to other countries remains unclear. Furthermore, the polynomial model may ignore the
      complicated inner interactions between multiple precursors, meteorological factors and ozone concentrations in the
      atmospheric environment (Chen et al., 2020), and may lead to large uncertainties. Consequently, ozone-precursors sensitivity,
      especially the transitional regime range across China, requires further in-depth analysis.

To this end, this research attempts to investigate the spatiotemporal variations of ozone formation regimes across China and
      identify the transitional regime range of $HCHO/NO_2$ based on the cross-verification of multiple models. Firstly, long-term
      variations of HCHO and $NO_2$ across China were analyzed. Next, the datasets of HCHO, $NO_2$ and ozone were examined
      using a polynomial model and a causality model respectively to reveal the crucial thresholds of $HCHO/NO_2$ that separates
      the $NO_x$-limited, VOCs-limited, and transitional regimes. Specifically, due to the large area of China and potential spatial
variations in ozone formation regimes, we respectively investigated ozone formation regimes in several major regions,
      including the North China Plain (NCP), Yangtze River delta (YRD), Pearl River delta (PRD), and Sichuan Basin (SCB) (The
      geographical locations of four megacity clusters were shown in Figure 1), to explore the spatiotemporal variations of ozone
      formation regimes. Meanwhile, we also compared the ozone formation regimes in urban and rural areas. This research sheds
      useful lights for better modeling complicated ozone-precursors relationship, understanding the major drivers for enhanced
ozone pollution and implementing specific emission-reduction measures to mitigate ozone pollution across China.

Figure 1 inserted here.

## 2 Materials and methods

### 2.1 Data sources

      In this study, OMI $HCHO/NO_2$ datasets were employed for exploring the spatiotemporal variations of HCHO and $NO_2$ in
China and calculating $HCHO/NO_2$. We connected surface ozone network data to HCHO, $NO_2$ and $HCHO/NO_2$, which
      served as the input data for running third-polynomial model and Convergent Cross Mapping (CCM). MODIS land cover
      product provided the spatial distribution of urban areas, which was employed for identifying urban and rural pixels.

### 2.1.1 OMI $HCHO/NO_2$

      Ozone Monitoring Instrument (OMI), on board the Aura satellite, monitors global solar backscatter in the UV/Vis domain
(270-500 nm). OMI provides daily global observations, which crosses the equator at 13:38 (local time) (Levelt et al., 2006).
      In this study, we employed daily level-3 gridded OMI HCHO product (OMHCHOd) from the Smith Astrophysical
      Observatory (SAO) (González Abad et al., 2015). The HCHO vertical columns are the weighted mean values for the 0.1° ×
      0.1° grid. Backscattered solar radiation, ranging from 328.5-356.5 nm, was used for fitting HCHO slant columns. Air mass
      factors (AMFs) were employed for converting HCHO slant columns to vertical columns (González Abad et al., 2015). The
validation report suggested that the error of this product was effectively controlled within 30% over polluted areas (González



Abad et al., 2015), and validated for detecting long-term variations of HCHO columns (Zhu et al., 2017; Shen et al., 2019b). The daily level-3 gridded OMI NO$_2$ product (OMNO2d), provided by NASA's Goddard Space Flight Center, were utilized in this study (Bucsela et al., 2013; Lamsal et al., 2014). The spatial resolution of OMNO2d is 0.25° and each grid is generated as the weighted average of the corresponding level-2 data pixels (Krotkov et al., 2017). Differential optical

absorption spectroscopy (DOAS) was employed for retrieving the NO$_2$ slant columns, which were successively transformed into tropospheric and stratospheric vertical columns through AMFs (Bucsela et al., 2013). The OMI NO$_2$ column product agrees well with other satellite products, and its overall uncertainties range from 30%-60% (Bucsela et al., 2013; Lamsal et al., 2014). To reduce uncertainties, we only selected those OMI HCHO and NO$_2$ data that (1) passed quality checks, (2) had a cloud coverage less than 30%, (3) had a solar zenith angle less than 60°, and (4) were not affected by row anomalies for

this study (Kroon et al., 2011; Zhu et al., 2014; Krotkov et al., 2017). The May-to-September OMI HCHO and NO$_2$ products were acquired from NASA's Goddard Earth Sciences Data and Information Services Center (https://disc.gsfc.nasa.gov/).

### 2.1.2 Surface ozone network data

The May-to-September hourly surface ozone concentrations from 2014 to 2019, were obtained from the China Ministry of Ecology and Environment (MEE) (https://quotsoft.net/air/). The unit of surface ozone concentrations in this dataset is μg/m$^3$.

The network had 1633 monitoring stations, which were distributed in 330 cities across China in 2019. We used the observation data from 13:00 to 14:00 at local time to match the overpass time of OMI. This dataset has been employed in many studies to investigate the variations of surface ozone concentrations in China (Li et al., 2019a; Shen et al., 2019a; Lu et al., 2020).

### 2.1.3 MODIS land cover product

The annual MODIS land cover product (MCD12C1) with a spatial resolution of 0.05° from 2005 to 2019 was employed for extracting urban and rural areas. The urban and water pixels from the International Geosphere-Biosphere Program (IGBP) classification layer were employed for the following processing. The land cover product was generated based on a decision tree algorithm with boosting techniques, and its overall accuracy was about 75% (Palmer et al., 2015; Bajocco et al., 2018). MCD12C1 product was obtained from NASA's Earth System Data and Information System (https://earthdata.nasa.gov/).

**2.1.4 Data pre-processing**

Due to different spatial resolution of OMI HCHO, OMI NO$_2$ and MCD12C1, bilinear interpolation method was used for resampling all above-mentioned products to the same spatial size (0.25° × 0.25°). Meanwhile, we also calculated mean hourly surface ozone concentrations on the 0.25° × 0.25° grid (Figure 1).



## 2.2 Methods

Chemical transport models, such as the global chemical transport model (GEOS-Chem) (Jin et al., 2017; Li et al., 2019a) and the Community Multiscale Air Quality Modeling System (CMAQ) (Duncan et al., 2010), have been frequently employed for exploring the ozone sensitivity to VOCs and $NO_x$. However, there were large biases in estimating the range of transitional regime based on chemical transport models (Jin et al., 2017; Jin et al., 2020), due to the uncertainties of emission inventory and the setting of model parameters. Employing observation data alone could effectively overcome these limitations, and the

relationships between ozone and its precursors were fitted using linear and polynomial models (Sun et al., 2018; Jin et al., 2020). Meanwhile, Convergent Cross Mapping (CCM) (Sugihara et al., 2012), as a robust causality analysis model, has been widely employed for quantifying the influences of meteorological factors on surface ozone and $PM_{2.5}$ concentrations (Chen et al., 2018; Chen et al., 2019b; Chen et al., 2020), which is a promising tool for investigating the relationships between ozone and its precursors. To increase the reliability of estimated range of transitional regime, both the polynomial model and

CCM were employed in this research. We employed the third-order polynomial model for fitting surface ozone concentrations to the indicator of $HCHO/NO_2$. CCM was employed for quantifying the influences of HCHO and $NO_2$ on surface ozone concentrations, and Wilcoxon test (Gehan, 1965) was used for examining whether the differences between the causality of HCHO and $NO_2$ on ozone concentrations at different ranges of $HCHO/NO_2$ was significant. Since the algorithms of the two models are quite different, their cross-verification provides useful reference for their reliability. Meanwhile,

Mann-Kendall (M-K) test (Kendall, 1948) was employed for exploring the spatiotemporal variations of HCHO, $NO_2$ and ozone formation regimes in China. Furthermore, we extracted all urban and rural areas in China and compared the differences of ozone formation regimes over these two types of areas. The workflow of the models employed in this study is shown in Figure 2.

Figure 2 inserted here.

### 2.2.1 Estimating the transitional range of ozone formation regime using polynomial simulation

HCHO and $NO_2$ are considered as proxies for VOCs and $NO_x$, respectively. $HCHO/NO_2$, as an effective indicator, has been widely employed for determining ozone formation regimes (Duncan et al., 2010; Jin and Holloway, 2015; Jin et al., 2017; Cheng et al., 2019; Jin et al., 2020). Pusede and Cohen (2012) suggested that ozone exceedance probability (OEP) was an effective indicator to interpret the ozone sensitivity to its precursors. The indicator is defined as the proportion of non-

attainment events (surface ozone concentrations exceeding 200 $\mu g/m^3$) in total events at a given range of $HCHO/NO_2$:

$$OEP = \frac{Events_{non-attainment}}{Events_{attainment} + Events_{non-attainment}} \tag{1}$$

where $Events_{attainment}$ and $Events_{non-attainment}$ denote the attainment and non-attainment events, respectively (Pusede and Cohen, 2012; Jin et al., 2020).



In this study, we used a third-order polynomial model (Jin et al., 2020) to explore the quantitative relationships between
HCHO/NO$_2$ and ozone exceedance probability. There were 174868 paired observations of surface ozone concentrations and
HCHO/NO$_2$ from 2014 to 2019. The peak of fitting curve highlights the turning point of VOC-limited and NO$_x$-limited
regimes (Jin et al., 2020). The range of HCHO/NO$_2$, which corresponded to the top 10% ozone exceedance probability, was
defined as the transitional regime.

### 2.2.2 Estimating the transitional range of ozone formation regime using Convergent Cross Mapping

We also employed Convergent cross mapping (CCM) (Sugihara et al., 2012), which could reduce the influences of other
factors such as meteorological conditions (Chen et al., 2019b; Chen et al., 2020), to extract the causal influences of HCHO
and NO$_2$ on surface ozone concentrations. Thanks to its capability of detecting weak coupling, CCM is advantageous for
reliably comparing the influences of different meteorological factors on surface ozone concentrations (Chen et al., 2020).
Therefore, we employed CCM to compare the sensitivity of ozone to HCHO and NO$_2$ at different ranges of HCHO/NO$_2$.
CCM utilizes convergent maps to demonstrate the bidirectional coupling between the time series of two variables. A
convergent curve indicates that one variable imposes influences on the other variable, whilst a non-convergent curve denotes
no causality between two variables. CCM calculates cross map skill ($\rho$ value) that explains the quantitative relationships.
Number of dimensions for the attractor reconstruction ($E$), time lag ($\tau$) and number of nearest neighbors to use for prediction
($b$) are required parameters for CCM. According to previous studies (Chen et al., 2020; Chen et al., 2019b), $E$, $\tau$ and $b$ was
set as 3, 2 and 4, respectively. Since the existence of missing values imposes negative impacts on CCM results, only the
consecutive time series were retained for this research. There were 1660 observation records of HCHO time series, NO$_2$ time
series and corresponding surface ozone time series. CCM was implemented using "pyEDM" package in Python. Wilcoxon
test (Gehan, 1965) was used to examine whether the differences of $\rho$ values between HCHO and NO$_2$ were significant at the
given HCHO/ NO$_2$. No significant difference was regarded as the transitional regime, while significant difference indicated
the VOC-limited or NO$_x$-limited regime.

### 2.2.3 Trend analysis

Mann-Kendall (M-K) (Kendall, 1948) test, which has been used in recent studies on HCHO and NO$_2$ (Cheng et al., 2019;
Wang et al., 2019; Zeb et al., 2019), was employed to estimate the significance of trends. M-K test is capable of processing
samples with random distributions and mitigating the effects of outliers. $Z$ value is calculated as follow:

$$Z = \begin{bmatrix} \dfrac{S-1}{\sqrt{Var(S)}}(S>0) \\ \dfrac{S+1}{\sqrt{Var(S)}}(S<0) \end{bmatrix} \qquad (2)$$

where $S$ denotes the statistic to be tested, $Var(S)$ stands for the variance of $S$.



The sign and absolute value of $Z$ indicate the direction and significance of trends, respectively. Specifically, the positive and negative values of $Z$ indicate the upward and downward trend. 1.28, 1.64 and 2.32 are the threshold values of $|Z|$, indicating the trends of samples pass the tests at 90%, 95% and 99%, respectively.

### 2.2.4 Comparison of ozone formation regimes in urban and rural areas in China

To compare the differences of ozone formation regimes in urban and rural areas in China, the key step is to extract urban and rural pixels, respectively. Urban pixels were used for buffer analysis (Imhoff et al., 2010) to identify rural pixels. Following Peng et al. (2018), two buffers were set for urban pixels to extract candidate rural pixels (Figure 3). We set the size of each buffer as 27.75 km, which was close to the size of the 0.25° × 0.25° grid (27.75 km ≈ 0.25°). The first and second buffers were determined as the urban fringes and candidate rural areas, respectively. Water pixels were firstly removed from candidate rural areas to avoid following uncertainties. Consequently, rural areas were regarded as buffers of 27.75-55.50 km surrounding urban areas. The use of two buffers not only assisted a complete separation of the urban and rural areas, but also minimized the uncertainties of meteorological conditions (Yao et al., 2019).

Figure 3 inserted here.

### 3 Results

#### 3.1 Spatial and temporal variations of HCHO and NO$_2$

Given the national Clean Air Action implemented in 2013, we set this year as a break point to explore the spatial and temporal variations of HCHO and NO$_2$ in 2005-2012 and 2013-2019, respectively. Figure 4 shows the spatial distribution of HCHO in the two periods. The mean HCHO values during the period of 2005-2012 and 2013-2019 were $4.335 \times 10^{15}$ molec/cm$^2$ and $4.845 \times 10^{15}$ molec/cm$^2$, characterized with a 12% increase. Both periods presented an increasing trends of HCHO, and the averaged value during the two periods were $0.164 \times 10^{15}$ molec/cm$^2$ year$^{-1}$ and $0.213 \times 10^{15}$ molec/cm$^2$ year$^{-1}$ (Figure 5). A faster increasing trend was detected during the period of 2013-2019. The variation trend of HCHO agreed well with previous studies (Jin and Holloway, 2015; Shen et al., 2019b). We also calculated the overall linear trends of HCHO in four megacity clusters from 2005 to 2019 (Figure 6). The largest and smallest increasing trends were shown in NCP and SCB, with a mean value of $0.136 \times 10^{15}$ molec/cm$^2$ year$^{-1}$ and $0.046 \times 10^{15}$ molec/cm$^2$ year$^{-1}$. The increasing trend of YRD and PRD were $0.066$ molec/cm$^2$ year$^{-1}$ and $0.058$ molec/cm$^2$ year$^{-1}$, respectively. Meanwhile, reversed trends were detected for NO$_2$ during the two periods (Figure 5), which was consistent with previous studies (Jin and Holloway, 2015; Li et al., 2019a). From 2005 to 2012, the averaged NO$_2$ was $2.027 \times 10^{15}$ molec/cm$^2$ and the annually mean increasing trend was $0.098 \times 10^{15}$ molec/cm$^2$ year$^{-1}$. Thanks to the implementation of Clean Air Action, the averaged NO$_2$ were reduced to 1.900 $\times 10^{15}$ molec/cm$^2$, with a decreasing trend of $-0.029 \times 10^{15}$ molec/cm$^2$ year$^{-1}$ from 2013 to 2019. Except for SCB, all other megacity clusters presented significant downward trends of NO2 from 2005 to 2019. Amongst these megacity clusters, NO$_2$





in YRD demonstrated the largest decreasing trend of $0.104 \times 10^{15}$ molec/cm$^2$ year$^{-1}$. NO$_2$ in NCP and PRD decreased by $0.010 \times 10^{15}$ molec/cm$^2$ year$^{-1}$ and $0.092 \times 10^{15}$ molec/cm$^2$ year$^{-1}$, respectively. A slightly increasing trend of $0.012 \times 10^{15}$ molec/cm$^2$ year$^{-1}$ was detected in SCB (Figure 7).


Figure 4 inserted here.

Figure 5 inserted here.

Figure 6 inserted here.

Figure 7 inserted here.

**3.2 Transitional range of ozone formation regime**

According to HCHO/NO$_2$, We divided the paired observations into 200 bins for the whole country and 100 bins for these megacity clusters, and the ozone exceedance probability was calculated for each bin. The third-order polynomial was employed for fitting ozone exceedance probability to HCHO/NO$_2$. As shown in Figure 8a, the peak of the fitting curve was 1.4, and the vertical shaded area indicated that the transitional regime over China ranged from 1.0 to 1.9. In addition to the regime range extracted at the national scale, we also examined the range of ozone formation regimes in four major megacity

clusters. The range of transitional regime for NCP, YRD, PRD and SCB was [1.2, 2.1], [1.0, 1.9], [0.9, 1.8] and [1.1, 2.0] respectively, which was generally consistent with the range at the national scale. The small differences between four megacity clusters across China suggested the range of transitional regime at the national scale [1.0, 1.9] can be employed to regional or local scale research, if small-scale data and investigation was not available.

Statistical bootstrapping was used for estimating the uncertainty of the fitting model. Specifically, we iteratively extracted 50

randomly selected subsets from the paired observations to run the model, and the uncertainty was defined as two standard deviations from the peak of the fitting curve. The uncertainty for the third-polynomial model was 0.4, indicating a significant nonlinear relationship between ozone exceedance probability and HCHO/NO$_2$.

Due to the limited data used for running CCM, we set the bin size of HCHO/NO$_2$ as 0.2 for collecting sufficient $\rho$ values to conduct Wilcoxon test. As shown in Figure 8b, there were no significant difference between $\rho$ of HCHO and NO$_2$ when

HCHO/NO$_2$ ranged from 0.9 to 1.9, which indirectly defined the range of transitional regime. For HCHO/NO$_2$ < 0.9, $\rho$ of HCHO was notably higher than that of NO$_2$, and this range was regarded as VOC-limited regime. Similarly, HCHO/NO$_2$ > 1.9 suggested the NO$_x$-limited regime. Through the cross verification, it was an important finding that the range of transitional ozone formation regime estimated using the third-order polynomial model and CCM was highly close, indicating the reliability of the extracted range.


Figure 8 inserted here.

**3.3 Ozone formation regimes in China**

NO$_2$ demonstrated a significant downward trend since 2013, while HCHO kept the increasing trend during the entire study period. Consequently, HCHO/NO$_2$ increased in a majority of regions across China. Specifically, the annually increasing



trend of HCHO/NO$_2$ in NCP, YRD and PRD was 0.035 year$^{-1}$, 0.023 year$^{-1}$ and 0.034 year$^{-1}$, respectively. Meanwhile, there

were no significant trends in SCB during this period (Figure 9). The variations of HCHO/NO$_2$ indicated the shrinkage of VOC-limited regime and the expansion of transitional and NO$_x$-limited regimes. Since the range of transitional regime estimated by third-order polynomial model and CCM was very close and the former included more reliable observation data, [1.0, 1.9] was employed for identifying different ozone formation regimes. In 2005, areas with the VOC-limited regime were concentrated in NCP, YRD and PRD. The proportions of areas with the VOC-limited regime in NCP, YRD and PRD were

26%, 16% and 6%, respectively. Areas with the transitional regime were mainly distributed in the marginal regions of those megacity clusters, and scatteredly distributed in SCB. Areas with the transitional regime occupied 60%, 50%, 14% and 20% in NCP, YRD, PRD and SCB. NO$_x$-limited regime dominated other areas (Figure 10a). In 2019, areas with the VOCs-limited regime decreased significantly, and was simply found in the fringe areas of NCP and YRD. The proportion of the VOCs-limited regime in NCP and YRD was 2% and 9%, respectively. The transitional regime was widely distributed NCP,

YRD and SCB, and occupied the 71 %, 56% and 36% of the total areas. The NO$_x$-limited regime still spread over a majority of China (Figure 10a). We calculated the annually mean $\rho$ of HCHO and NO$_2$ over those megacity clusters from 2014 to 2019 (Figure 10b). For all megacity clusters, the $\rho$ of NO$_2$ was higher than HCHO, indicating that NO$_2$ was the dominant factor for surface ozone concentrations. Both models suggested that NO$_2$ played a more important role in affecting surface ozone concentrations than HCHO. In the past several years, NO$_x$-oriented emission-reduction has been conducted across

China, leading to the continuous decrease of NO$_x$ concentrations. Since both VOCs and NO$_x$ imposed positive influences on surface ozone concentrations under the transitional and NO$_x$-limited ozone formation regime, the upward trend of HCHO across China might explain recent soaring ozone concentrations across China (Shen et al., 2019a; Lu et al., 2020). It is noted that the difference between the $\rho$ of NO$_2$ and HCHO decreased notably in NCP and YRD. This may be attributed to the following reason. NCP and YRD are the regions that received severe PM$_{2.5}$ pollution and strict NO$_x$-reduction policies have

been conducted since 2013. With the remarkably reduced NO$_2$ concentrations, the variations of HCHO concentrations plays an increasingly important role in affecting ozone concentrations in NCP and YRD. The reduction of VOCs emissions is key for an effective management of surface ozone pollution in NCP and YRP.

Figure 9 inserted here.

Figure 10 inserted here.

**3.4 Variations of ozone formation regimes in urban and rural areas**

Previous studies suggested that the differences of ozone formation regimes existed between urban and rural areas (Tong et al., 2017; Liu et al., 2018b; Cheng et al., 2019). We extracted HCHO and NO$_2$ columns in urban and rural pixels in those megacity clusters, and calculated the annually averaged HCHO/NO$_2$ (Figure 11). For NCP, HCHO/NO$_2$ in urban areas was higher than 1.0 since 2015, indicating a transformation from VOC-limited to transitional regime. The increase of

HCHO/NO$_2$ was attributed to the reversed variation trends of HCHO and NO$_2$. The rising HCHO resulted from the increase of anthropogenic emissions and biogenic volatile organic compound (BVOC) (Shen et al., 2019b; Wang et al., 2020), while



the implementation of Clean Air Action imposed notable influences on the decrease of $NO_2$ (Chen et al., 2019a). $HCHO/NO_2$ in rural areas was in the range of [1.0, 1.9], indicating rural areas were occupied by the transitional regime from 2005 to 2019. For YRD, which was occupied by the transitional regime, no variation of ozone formation regime was found

in urban areas. In rural areas, $HCHO/NO_2$ temporally exceeded the threshold of 1.9 from 2016 to 2018, indicating the ozone formation regime changed from transitional to $NO_x$-limited. This phenomenon was attributed to the slight decline of HCHO, which might be attributed to the restrictions on crop residue burning in this area (Zhuang et al., 2018; Shen et al., 2019b). Due to the large differences of $NO_2$ concentrations, the urban and rural areas in PRD was dominated by transitional regime and $NO_x$-limited regime. For SCB, $HCHO/NO_2$ in both urban and rural areas fluctuated around the threshold value of 1.9,

and no significant difference between urban and rural areas was found.

Figure 11 inserted here.

## 4 Discussion

This research employed CCM and third-order polynomial model to estimate the transitional regime of ozone formation across China and the calculated range of $HCHO/NO_2$ was [0.9, 1.9] and [1.0, 1.9], respectively. Given the lack of actual

reference data, the close output from two different models provides a reliable reference for better understanding ozone formation regimes. Our findings were generally consistent with previous studies. For US, Duncan et al. (2010) and Choi et al. (2012) employed the OMI and GOME-2 data, whose 0.25° resolution was close to this research, and calculated the range of transitional regime as [1.0, 2.0]. The similar range of transitional regime in US and China further proved the reliability of the calculated range [1.0, 1.9] at a national scale. On the other side, the range of transitional regime can vary significantly across

regions (Schroeder et al., 2017; Jin et al., 2020). Sun et al. (2018) employed station-based data and calculated the range of transitional regime in Anhui Province, China as [1.3, 2.8], which was notably higher than the range across China. Jin et al. (2020) calculated the range of transitional regime in several major regions in the US using QA4ECV dataset, whose spatial resolution was 0.125°, and the output [3.2, 4.1] was much larger than the averaged range of transitional regime across US. One reason could be the severe ozone pollution in mega cities, leading to different ranges of transitional regime. Meanwhile,

the calculated range of transitional regime is closely related to the spatial resolution of employed HCHO and NO2 data, and high-resolution data are more advantageous in extracting the sensitivity of ozone concentrations to precursors at the local scale (Martin et al., 2004; Jin et al., 2017; Jin et al., 2020). In addition to the locations and the spatial resolution of data sources, the uncertainties of OMI HCHO and $NO_2$ datasets might impose negative influences on the estimation of transitional regime range (Duncan et al., 2010; Jin et al., 2017; Schroeder et al., 2017; Jin et al., 2020). Firstly, errors exist in

the retrieval of HCHO and $NO_2$ vertical columns. Secondly, vertical mixing was not homogeneous, weakening the capability of using HCHO and $NO_2$ vertical columns to explore the near-surface ozone-precursors sensitivity. Therefore, future improvement of earth observation techniques and the spatiotemporal resolution of HCHO and $NO_2$ products can further enhance the accuracy of the estimated range of transitional regime. In general, according to the cross-verification and





comparison with previous studies, [1.0, 1.9] from this research is a reliable range for transitional ozone formation regime
across China and can be used as an approximate criterion to follow when implementing national emission-reduction policies.
On the other hand, given the potential variations of transitional regimes in different regions, when conducting small-scale
research, the range of [1.0, 1.9] may be adapted accordingly based on local data.

Previous studies on the range of ozone formation regimes were mainly conducted using statistical models or chemical
transport models. For this research, we employed both a statistical and a causality models to cross-verify the range of
transitional regimes. Despite a relatively high fitting accuracy in terms of uncertainties, the findings from these studies could
not be effectively compared or interpreted, due to the lack of reliable reference data. To this end, as well as numerical
models, lab experiments should also be considered to extract more precise description of ozone-precursors relationship. With
the rapid development of atmospheric science, smog chambers have been increasingly employed to investigate complicated
interactions between multiple precursors. By setting specific meteorological conditions (e.g. temperature and humidity) and
gradually adjusting the proportion of different precursors, how the proportion of $NO_2$ and HCHO affect ozone formation
regime can be better explained in a theoretical environment. With more reliable experimental reference data, the model-
based analysis on the range of transitional regime at local, regional and national scale can be further improved accordingly.

According to the temporal variations of OMI $NO_2$ concentrations across China, a notable decreasing trend was observed in
three major megacity clusters, NCP, YRD and PRD. These regions were heavily polluted by $PM_{2.5}$ and the notable decrease
of $NO_2$ was mainly attributed to the national Clean Air Action since 2013 (Zheng et al., 2018), which aimed to reduce $PM_{2.5}$
concentrations by cutting $NO_x$ emissions. Conversely, HCHO concentrations during this period increased remarkably across
China, due to the combined effects of anthropogenic and biogenetic emissions (Shen et al., 2019b; Wang et al., 2020). The
distinct temporal variations of $NO_2$ and HCHO led to the increase of $HCHO/NO_2$, and the increase of transitional areas and
$NO_x$-limited regime areas. From 2013-2019, all these regions were dominated by the transitional or $NO_x$-limited regimes.
Attributed to the long-term variation of formation regimes, a more complicated and fragmented spatial pattern was observed
across China. Consequently, for an effective control of ozone pollution, the emission-reduction of both $NO_x$ and VOCs is
required. Especially for NCP and YRD, where the $NO_x$ has been remarkably reduced, effective approaches for controlling
VOCs emissions are essential for preventing ozone pollution. This finding was consistent with previous studies (Li et al.,
2019b), which recommended the simultaneous reduction of $NO_x$ and VOCs for mitigating the composite airborne pollution
in China. Admittedly, compared with $NO_x$-reduction, the VOCs-reduction is more complicated and the output of
anthropogenic VOCs reduction is more unpredictable. In this case, reducing biogenic VOCs emissions can also be a
potential solution. VOCs emitted by vegetation takes up to 50% of total VOCs in the atmospheric environment, especially in
summer. The key factor that may cause enhanced biogenic emissions is summertime high temperature in summer (Chen et
al., 2020). Therefore, such projects as wind corridors or contingent artificial precipitation, which can effectively reduce
urban heat effects, should be implemented properly to avoid summertime heat waves and successive ozone pollution (e.g.
summer, 2017).



The large spatial variations of HCHO/NO$_2$, especially the rapid increase of transitional regime areas across China, indicates that a unified NO$_x$-VOCs reduction strategy is not feasible for the entire country. Instead, to effectively reduce ozone concentrations, the specific proportion of NO$_x$ and VOCs reduction should be carefully set according to local HCHO/NO$_2$.

Meanwhile, due to the large differences in vehicle and industrial emissions (Cheng et al., 2019), the concentration of NO$_x$ is notably higher in urban areas. Therefore, the further reduction of NO$_x$ emissions exerts a stronger influence on ozone reduction in rural areas compared to urban areas.

## 5 Conclusions

To better understand the spatiotemporal variations of ozone formation regimes across China, we employed the third-order

polynomial model and CCM to estimate the range of transitional regime from 2005 to 2019, the results of which were [1.0, 1.9] and [0.9, 1.9], respectively. The close outputs from two distinct models proved the reliability of the extracted range. At the regional scale, we also investigated the range of transitional regime in four megacity clusters and found the range in NCP, YRD, PRD and SCB demonstrated limited differences and was generally consistent with the range at the national scale. The reverse trends of HCHO and NO$_2$ led to the increase of HCHO/NO$_2$, indicating China was dominated by the transitional and

NO$_x$-limited regimes in recent years. We also found that the $\rho$ of NO$_2$ was higher than HCHO at all megacities, suggesting that the reduction of NO$_x$ emissions would become more effective in controlling surface ozone concentrations. Meanwhile, given the rising VOCs emissions, the simultaneous reduction of NO$_x$ and VOCs would be more effective than the sole reduction of NO$_x$ in mitigating ozone pollution. Finally, the comparison of ozone regimes in urban and rural areas suggested that the reduction of NO$_x$ emissions would impose stronger impacts on the control of ozone pollution in rural areas.

**Author contributions**

Ruiyuan Li: Writing – original draft, Visualization, Formal analysis. Miaoqing Xu: Data curation, Visualization, Formal analysis. Manchun Li: Conceptualization, Methodology. Ziyue Chen: Wrting – original draft, Conceptualization, Formal analysis. Bingbo Gao: Data curation, Formal analysis. Na Zhao: Data curation, Visualization. Qi Yao: Data curation, Visualization.

**Competing interests**

The authors declare that they have no conflict of interest.



**Acknowlegements**

This research is supported by the Beijing Natural Science Foundation (Grant No. 8202031), Open Fund of the State Key Laboratory of Remote Sensing Science (Grant No. OFSLRSS201926), the Open Fund of the State Key Laboratory of Resources and Environmental Information System.

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



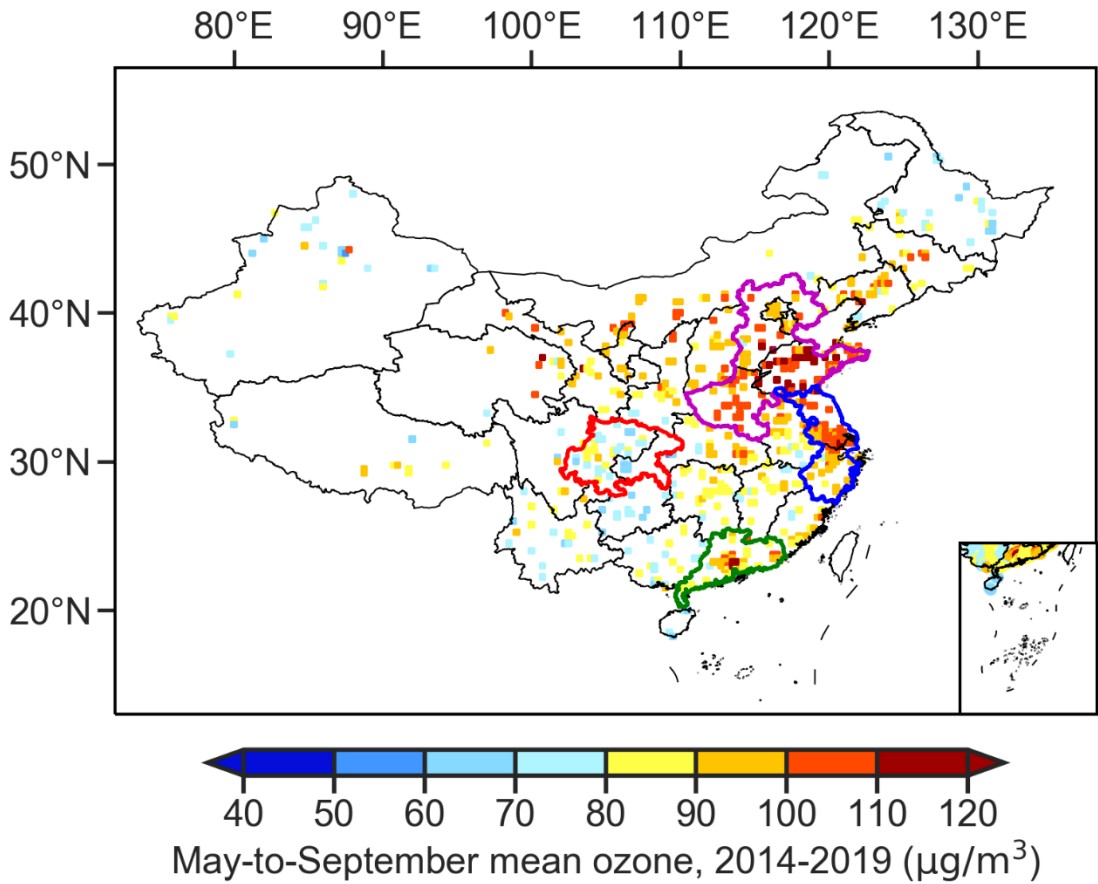

**Figure 1: The May-to-September mean hourly surface ozone network data from 2014 to 2019. Mean hourly surface ozone concentrations were calculated on the 0.25° × 0.25° grid. Purple, blue, green and red outlines indicate the boundaries of North China Plain (NCP), Yangtze River delta (YRD), Pearl River delta (PRD), and Sichuan Basin (SCB), respectively.**




**Figure 2: The workflow of the polynomial simulation and the causality analysis.**





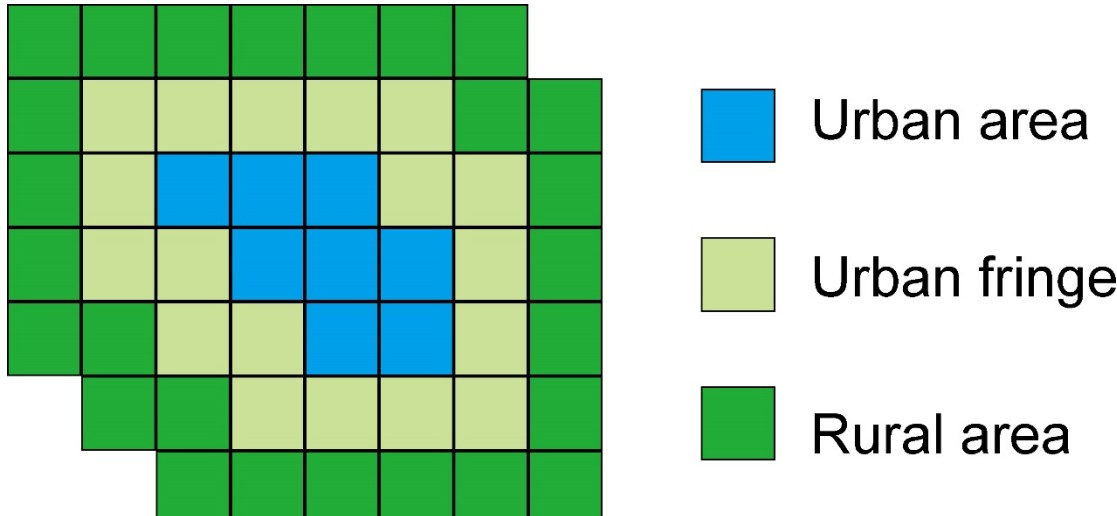

**Figure 3: The geographical locations of urban area, urban fringe and rural area.**



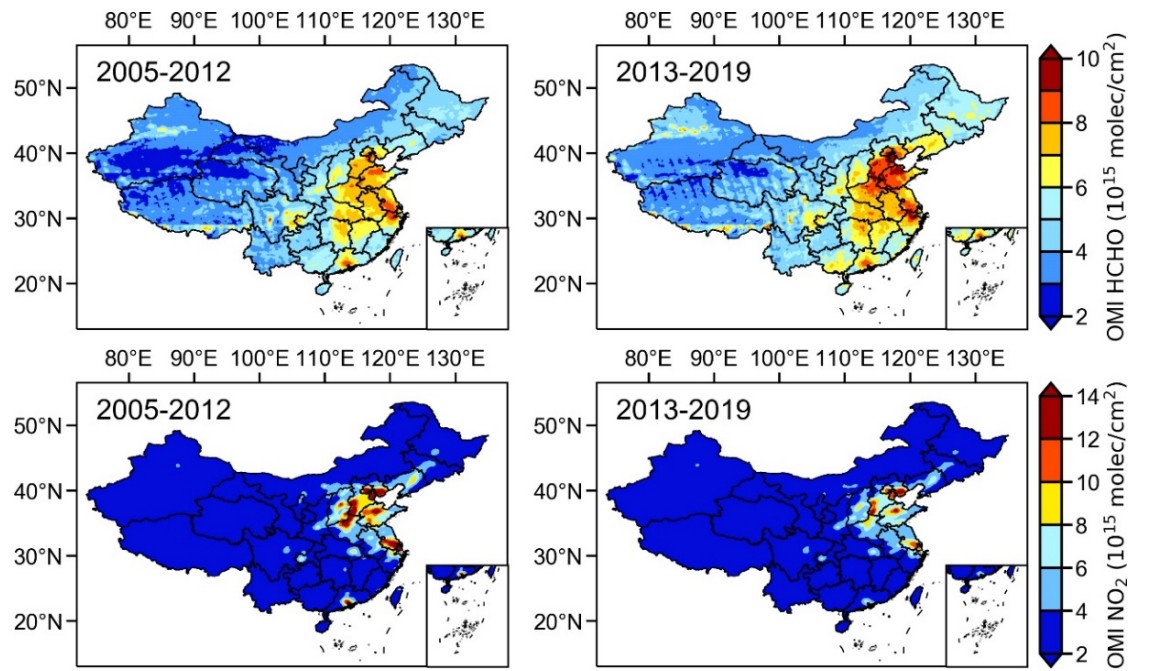

520       **Figure 4: May-to-September averaged HCHO and NO2 across China during the period of 2005-2012 and 2013-2019.**





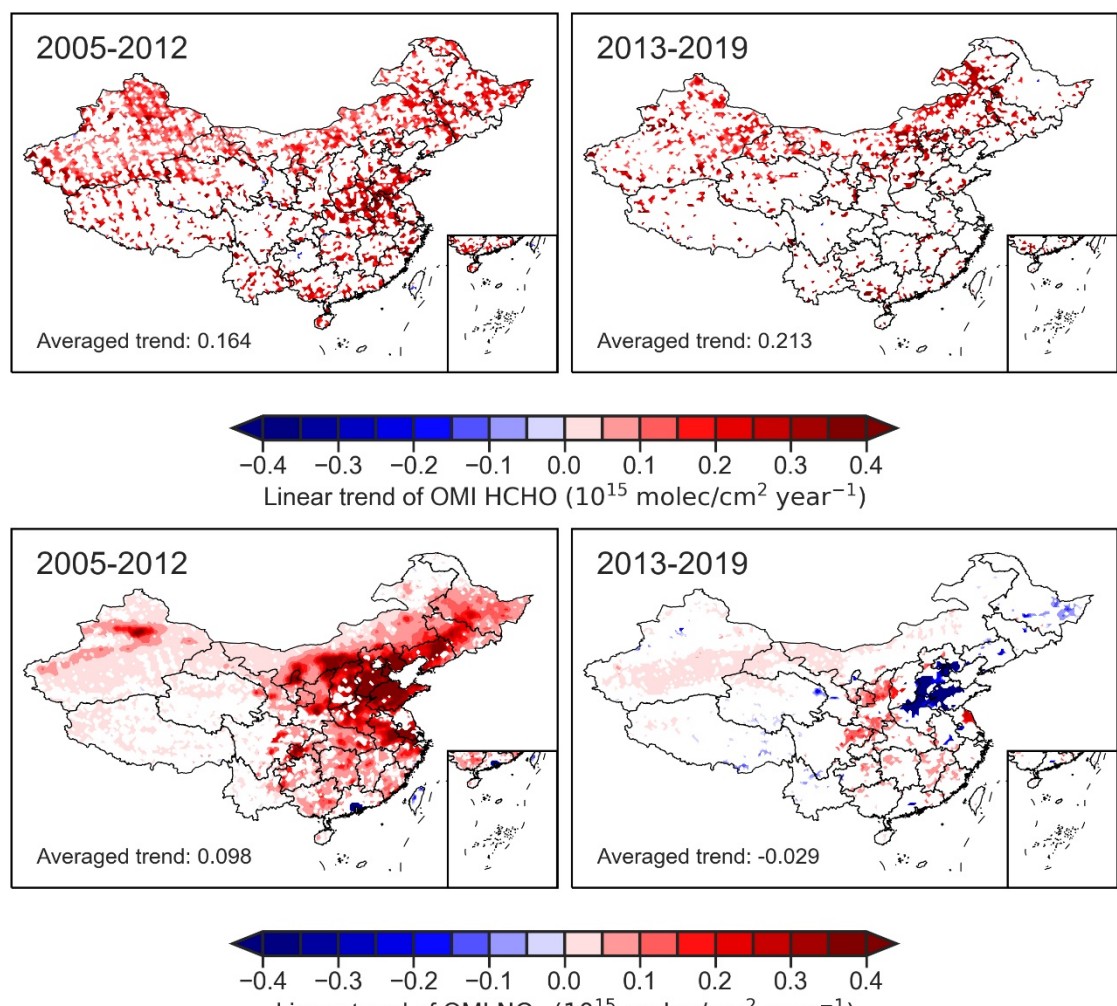

**Figure 5: The linear trends of May-to-September HCHO and NO₂ across China during the period of 2005-2012 and 2013-2019.**

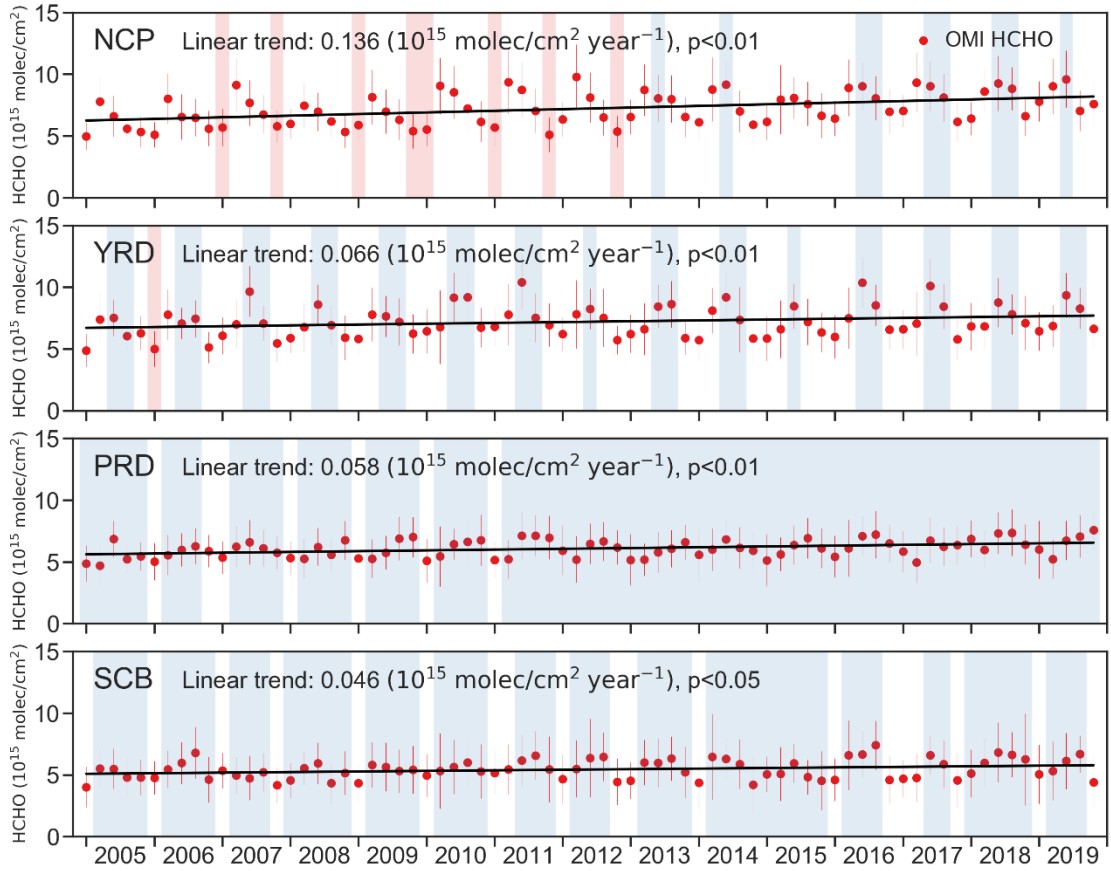

**Figure 6: The time series of HCHO columns in the four megacity clusters from 2005 to 2019. Black lines indicate the linear trend**
**of HCHO columns. Red, white and blue areas stand for VOC-limited, transitional and NO$_x$-limited regimes, respectively.**

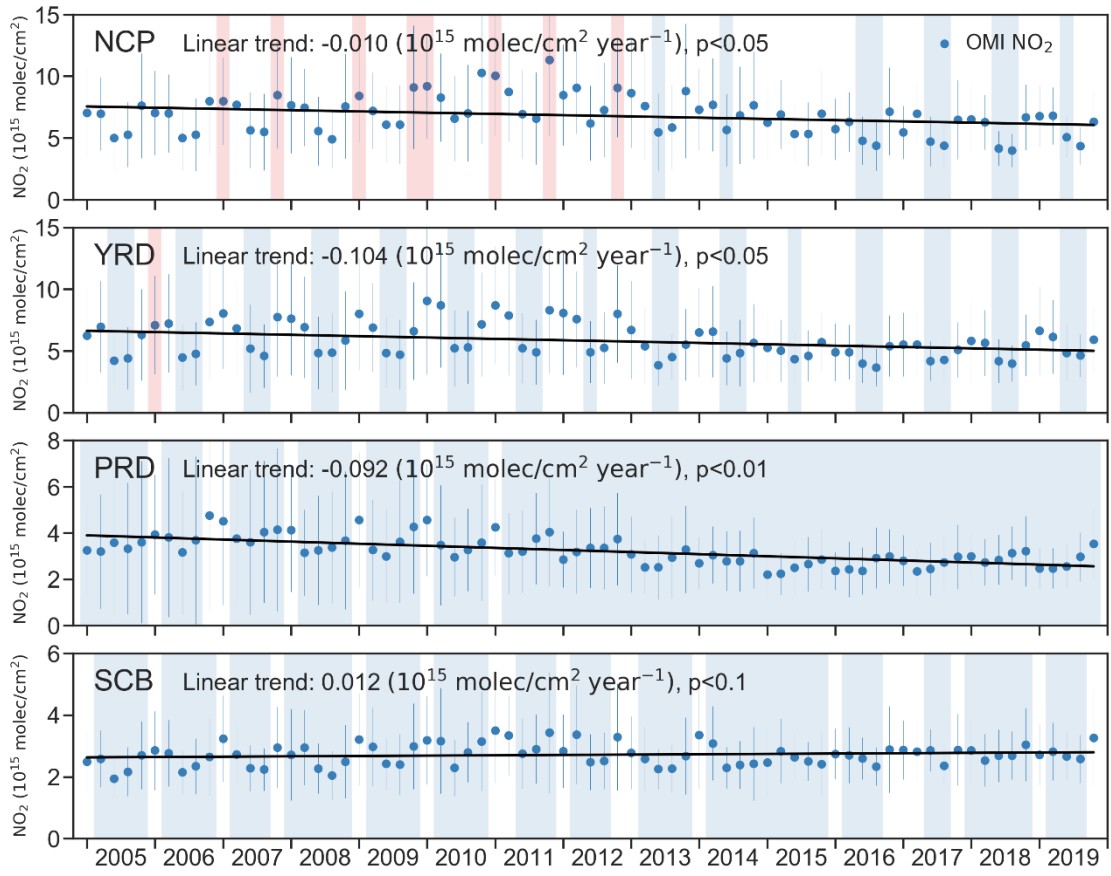

**Figure 7: The time series of NO₂ columns in the four megacity clusters from 2005 to 2019. Black lines indicate the linear trend of NO₂ columns. Red, white and blue areas stand for VOC-limited, transitional and NOₓ-limited regimes, respectively.**





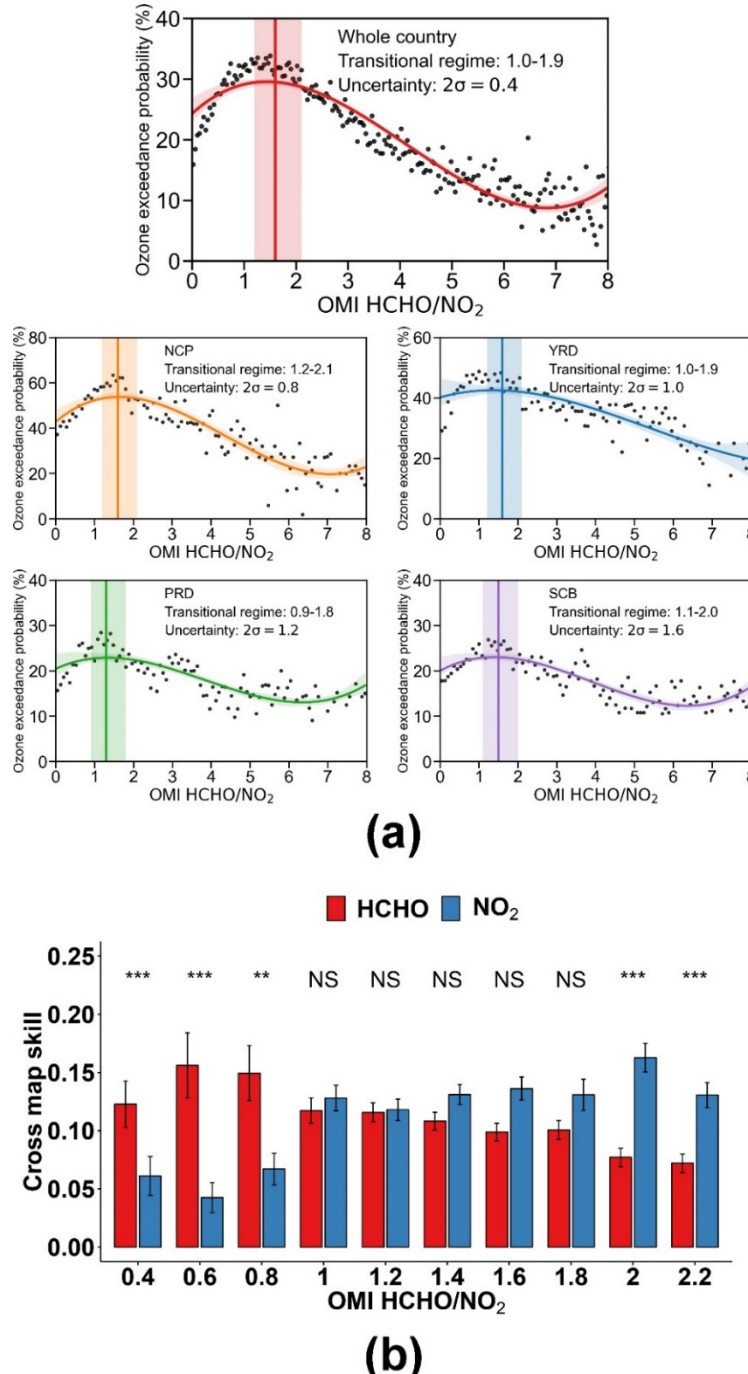


**Figure 8: (a) Fitting ozone exceedance probability to HCHO/NO₂ through third-order polynomial model. The curve indicated the fitting result of third-order polynomial. The vertical line denoted the maximum of curve and the shaded area represented the top 10% ozone exceedance probability. (b) The cross map skill of HCHO and NO₂ on surface ozone (The skill of using HCHO and NO₂ for predicting surface ozone concentrations) at different ranges of HCHO/NO₂. The symbols and texts above the bars were**
**the results of Wilcoxon test. ***, ** indicated the difference was significant at p = 0.01, 0.05 confidence level, respectively. NS suggested non-significant differences.**

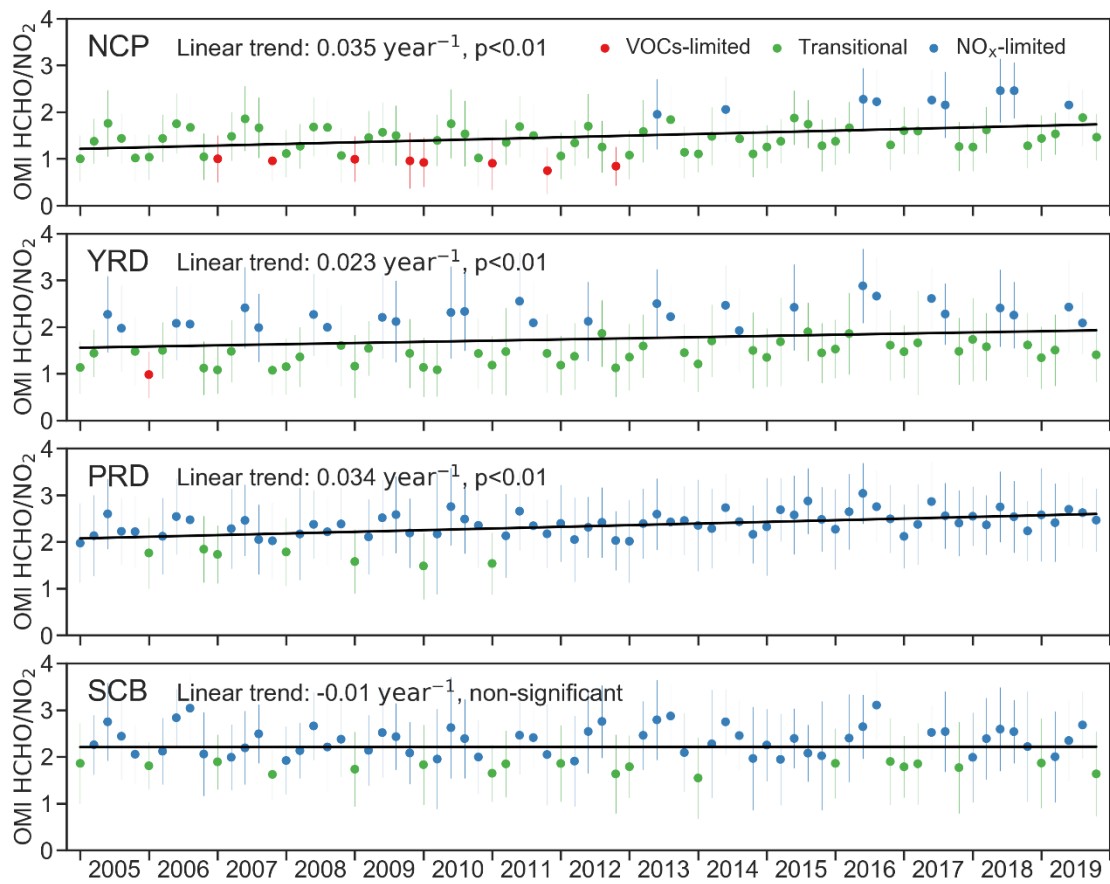

**Figure 9: The time series of HCHO/NO₂ in the four megacity clusters from 2005 to 2019. Black lines indicate the linear trend of HCHO/NO₂. Red, green and blue dots stand for VOC-limited, transitional and NOₓ-limited regimes, respectively.**



**Figure 10:** (a) The spatial distribution of HCHO/NO₂ across China in 2005 and 2019. The boundaries of NCP, YRD, PRD and SCV are denoted with the purple, blue, yellow and red bold lines. Red, green and blue stand for VOC-limited, transitional and NO$_x$-limited regimes. (b) The annual mean cross map skill ($\rho$ value) of four megacity clusters. The red and blue shadow areas indicate the standard deviations.




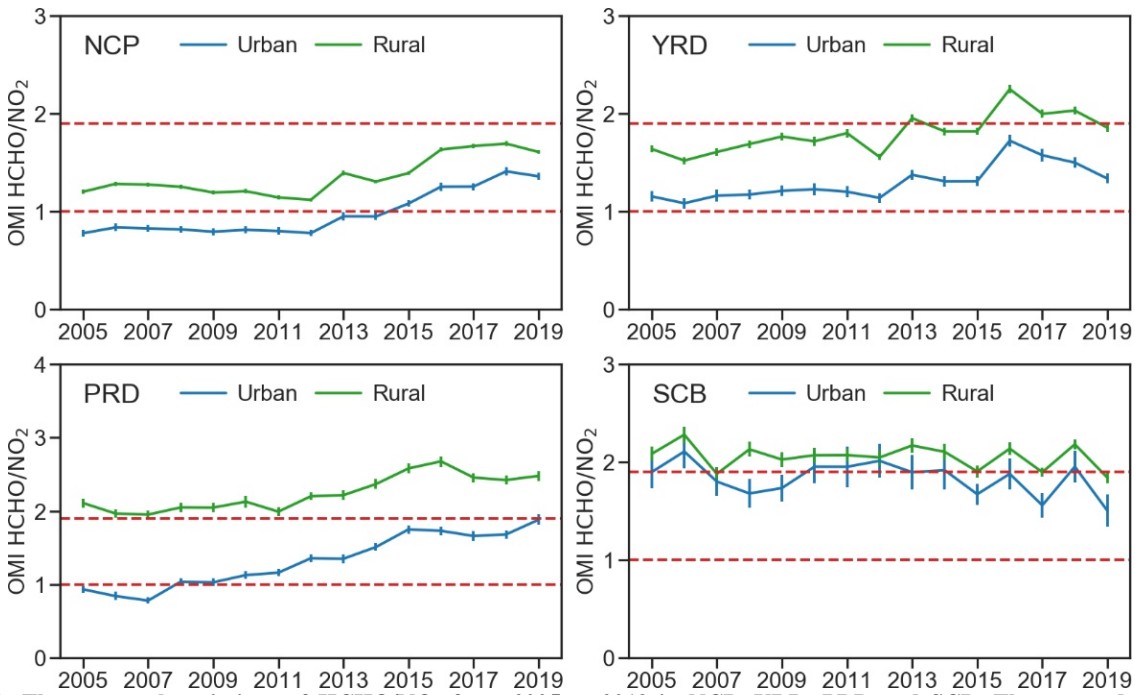

**Figure 11: The temporal variations of HCHO/NO₂ from 2005 to 2019 in NCP, YRD, PRD and SCB. The two red dash lines indicate the threshold values of 1.0 and 1.9, which separate the NO$_x$-limited, transitional and VOC-limited regime.**