# Peer review of "Identifying the spatiotemporal variations of ozone formation regimes across China from 2005 to 2019 based on polynomial simulation and causality analysis"

_Atmospheric Chemistry and Physics, 2021_

## Author Comment (AC4)

**To reviewer 1**

**Thanks so much for your valuable comments, which helped us so much revise the manuscript. We have fully revised it according to your comments. We are more than willing to conduct further revisions if additional requirements are given. The responses to the questions raised in your comment are as follows.**

**In addition, we are very sorry that we posted wrong versions of author comment before.**

1. Measure and attribute spatial stratified heterogeneity (SSH).

**Response: Thanks so much for this comment. We employed the geographical detector to measure spatial stratified heterogeneity (SSH). Population and strata were annual surface ozone concentrations and the boundaries of four megacity clusters, respectively. Furthermore, we attributed SSH according to the methodology and findings from previous studies.**

**We added the description of the geographical detector at lines 159-168 of the revised manuscript:**

**"Since we aimed to apply a global model to determine the transitional range, it was necessary to examine whether the surface ozone concentrations in China was of spatial stratified heterogeneity (SSH), as suggested by Wang et al. (2016). We employed the geographical detector (Wang et al. 2010) to measure the SSH of surface ozone concentrations. The geographical detector calculates q-statistic to quantify SSH and the equation is summarized as follows:**

$$q = 1 - \frac{\sum_{h=1}^{L} N_h \sigma_h^2}{N \sigma^2}$$

**Where $N$ and $\sigma^2$ denote the number of samples and the variance of population, $h$ is the number of stratifications. The range of q-statistic is [0, 1]. The larger the q-statistic is, the stronger the SSH is. In this study, the boundaries of four megacity clusters served as strata. If the SSH is detected based on above-mentioned stratification, we could apply the polynomial model in each strata, separately."**

**The results of the geographical detector were shown in Table 1.**

**Table 1: The q-statistic and p-value calculated by the geographical detector, which indicate the SSH of annual May-to-September mean surface ozone concentrations in China.  *, ** and *** of p-value indicate statistical significance at $\alpha$ = 0.05, 0.01 and < 0.001 level, respectively.**

| Year | q-statistic | p-value |
|------|-------------|---------|
| 2014 | 0.295*** | $9.621 \times 10^{-10}$ |
| 2015 | 0.325*** | $8.059 \times 10^{-10}$ |
| 2016 | 0.366*** | $4.803 \times 10^{-10}$ |
| 2017 | 0.609*** | $9.975 \times 10^{-10}$ |
| 2018 | 0.512*** | $2.647 \times 10^{-10}$ |
| 2019 | 0.708*** | $2.199 \times 10^{-10}$ |

**We attributed SSH at lines 241-245 of the revised version:**

**"As suggested by Chen et al. (2020), meteorological factors including temperature, humidity and sunshine duration imposed great impacts on surface ozone concentration. Moreover, the composition of ozone precursors was closely related to ozone levels (Cheng et al., 2019). Both the meteorological conditions and ozone precursors contributed to the SSH of surface ozone concentrations across China."**

**Chen, Z., Li, R., Chen, D., Zhuang, Y., Gao, B., Yang, L., Li, M.: Understanding the causal influence of major meteorological factors on ground ozone concentrations across China, J. Clean Prod., 242, doi: 10.1016/j.jclepro.2019.118498, 2020.**

**Cheng, N., Li, R., Xu, C., Chen, Z., Chen, D., Meng, F., Cheng, B., Ma, Z., Zhuang, Y., He, B., Gao, B.: Ground ozone variations at an urban and a rural station in Beijing from 2006 to 2017: Trend, meteorological influences and formation regimes, J. Clean Prod., 235, 11-20, doi: 10.1016/j.jclepro.2019.06.204, 2019.**

**Wang, J., Li, X., Christakos, G., Liao, Y., Zhang, T., Gu, X., and Zheng, X.: Geographical detectors-based health risk assessment and its application in the neural tube defects study of the Heshun region, China, Int. J. Geogr. Inf. Sci., 24, 107-127, doi: 10.1080/13658810802443457, 2010.**

**Wang, J., Zhang, T., and Fu, B.: A measure of spatial stratified heterogeneity, Ecol. Indic., 67, 250-256, doi: 10.1016/j.ecolind.2016.02.052, 2016.**

2. If the population is SSH and not all strata are sampled, the sample is biased; a global model would be confounded if the population is SSH.

**Response: Thanks so much for pointing this out. Following the results of the geographical detector, all the annual surface ozone concentrations during the period of 2014-2019 were SSH. Therefore, it was necessary to apply the polynomial model in each strata, and the fitting results were shown in Figure 8a. Thanks again for this valuable suggestions, which improved the manuscript significantly.**

[Figure]

**Figure 8: (a) Fitting ozone exceedance probability to HCHO/NO₂ through third-order polynomial model. The curve indicates the fitting result of third-order polynomial. The vertical line denotes the maximum of curve and the shaded area represents the top 10% ozone exceedance probability. (b) The cross map skill of HCHO and NO₂ on surface ozone (The skill of using HCHO and NO₂ for predicting surface ozone concentrations) at different ranges of HCHO/NO₂. The symbols and texts above the bars are the results of Wilcoxon test. \*\*\*, \*\* indicate the difference was significant at p = 0.01, 0.05 confidence level, respectively. NS suggestes non-significant differences.**

3. Provide the main equations of your model.

**Response: Thanks so much for this comment. We added the main equations of CCM at lines 177-183 of the revised manuscript.**

**"The main idea of CCM is summarized as follows. Firstly, CCM defines {X} and {Y} as the temporal variations of two variables X and Y. {X} generates the shadow manifold $M_X$. Following this, the location of lagged-coordinate vector on $M_X$, x(t) is determined, and then E + 1 nearest neighboring points of x(t) are extracted. Finally, the cross-mapped estimate of Y(t), $Y(t)|M_X$ are calculated as follows:**

$$Y(t)|M_X = \sum_{i=1}^{E+1} \omega_i Y(t_i)$$

**Where $\omega_i$ stands for a weight calculated based on the distance between X(t) and its ith nearest neighboring point. $Y(t_i)$ stands for contemporaneous value of Y."**

4. Justify your approach is the best one, considering many alternatives.

**Response: Thanks so much for these valuable comments. At line 126-130, we provided the limitations of chemical transport models. Meanwhile, we added the advantages of the models employed in this research at lines 325-329.**

**"First, only a few parameters are required for polynomial model and CCM, which effectively reduced the uncertainties of model setting. Second, considering the differences between model and satellite retrieved datasets (Jin et al., 2020), only observation data were employed in this research, which reduced potential data inconsistences and uncertainties. Most importantly, given the lack of actual reference data, this research employed two different models to examine ozone formation regimes and the close outputs further proved the reliability of this research."**

**Jin, X., Fiore, A., Boersma, K.F., De Smedt, I., Valin, L.: Inferring changes in summertime surface ozone-NO$_x$-VOC chemistry over U.S. urban areas from two decades of satellite and ground-based observations, Environ. Sci. Technol., 54, 11, 6518-6529, doi: 10.1021/acs.est.9b07785, 2020.**

5. Justify your approach and findings are "causality".

**Response: We are very sorry that we did not clearly describe the causality model, Convergent Cross Mapping (CCM). CCM utilizes convergent maps to demonstrate the**

bidirectional coupling between the time series of two variables. A convergent curve indicates that one variable imposes influences on the other variable, whilst a non-convergent curve denotes no causality between two variables. CCM calculates cross map skill ($\rho$ value) that explains the quantitative relationships. In this research, we compared the differences of $\rho$ values between HCHO and $NO_2$ at the given range of HCHO/ $NO_2$ to estimate the transitional regime range.

---

## Author Comment (AC5)

**To reviewer 2**

**Thanks so much for your valuable comments, which helped us so much revise the manuscript. We have fully revised it according to all your general and detailed comments. We are more than willing to conduct further revisions if additional requirements are given. The responses to the questions raised in your comment are as follows.**

1. The novel insights and research contribution of this study should be better articulated. From the results and discussion section, I found several statements "Page 7, lines 202-203: the variation trend of HCHO agreed well with previous studies (Jin and Holloway, 2015; Shen et al., 2019b)", "Page 7, lines 207-208: which was consistent with previous studies (Jin and Holloway, 2015; Li et al., 2019a)", "Page 10, line 291: our findings were generally consistent with previous studies". It would be better to highlight the position of this study and further justify the research advance, e.g., regarding the methods and datasets, or a more comprehensive picture of ozone formation regimes in China.

**Response: Thanks so much for these valuable comments. The comparison of previous studies and this research mainly aimed to add reliability of estimated transitional regime range. But your comment is quite right and we should highlight the unique contribution and findings of this research. In the revised manuscript, the advances of this research was added at lines 324-329 of the revised version.**

**"In addition to the generally consistent outputs, some advances of this research are listed as follows. First, only a few parameters are required for polynomial model and CCM, which effectively reduced the uncertainties of model setting. Second, considering the differences between model and satellite retrieved datasets (Jin et al., 2020), only observation data were employed in this research, which reduced potential data inconsistences and uncertainties. Most importantly, given the lack of actual reference data, this research employed two different models to examine ozone formation regimes and the close outputs further proved the reliability of this research."**

**Thanks again for your valuable comments, which improved the manuscript significantly.**

2. The authors set the implementation of Clean Air Action in 2013 as the breakpoint, which is appropriate for the comparison between these two baselines. However, the contribution from this

policy to driving the decreased NO$_2$ should be acknowledged in a more systematic and quantitative way, by adjusting a number of confounding factors.

**Response: Thanks so much for pointing this out. Yes, the implementation of clean air action has exerted a strong influence on the reduction of NO$_x$ across China, and the effects of clean air action have been massively studied in recent studies. Since the aim of this research was to estimate the transitional range of ozone formation regime and the impacts of Clean Air Action was widely discussed by previous studies, we added relevant explanations at lines 357-362 of the revised manuscript s:**

**"The influence of Clean Air Action on the reduction of PM$_{2.5}$ concentrations and NO$_x$ has been investigated by previous studies. Zheng et al. (2018) employed index decomposition analysis to quantify the contribution of the Clean Air Action, and suggested that the decreasing rate of NO$_x$ significantly accelerated since 2013. Moreover, Zhang et al. (2020) employed random forest algorithm to remove the effects of meteorological conditions, and evaluated the impacts of Clean Air Action. The results demonstrated that the deweathered NO$_2$ concentrations in winter 2007 and 2017 were 70.3 μg/m$^3$ and 59.1 μg/m$^3$, with a decreasing rate of 16%."**

**Thanks again for this valuable comment.**

3. The scaling biases between the station-based observations (i.e., point) and remote sensing based measurement (i.e., 0.25-degree footprint) should be discussed, especially for heterogeneous land cover/land uses.

**Response: This is a very good point. According to your comment, we discussed the potential scaling biases at lines 330-333 at the revised version.**

**"First, the accuracy of the estimated range of transitional regime might be influenced by the scaling biases between station-based observations of surface ozone and space-based HCHO and NO$_2$. Since ozone monitoring stations are mainly distributed in urban areas, and a 0.25 $^\circ$ × 0.25 $^\circ$ grid might cover both the urban and rural areas, the surface ozone concentrations of a grid may be overestimated."**

**Thanks again for this valuable comment.**